# Diet drove brain and dental morphological coevolution in strepsirrhine primates

**Camilo López-Aguirre** ⓘ *, **Madlen M. Lang, Mary T. Silcox**

Department of Anthropology, University of Toronto Scarborough, Toronto, Ontario, Canada

* c.lopezaguirre@utoronto.ca

**Data Availability Statement:** All raw data and code are available in the Supporting information files.

**Funding:** Funding for this project was provided by the University of Toronto Scarborough Office of the

## Abstract

The evolution of the remarkably complex primate brain has been a topic of great interest for decades. Multiple factors have been proposed to explain the comparatively larger primate brain (relative to body mass), with recent studies indicating diet has the greatest explanatory power. Dietary specialisations also correlate with dental adaptations, providing a potential evolutionary link between brain and dental morphological evolution. However, unambiguous evidence of association between brain and dental phenotypes in primates remains elusive. Here we investigate the effect of diet on variation in primate brain and dental morphology and test whether the two anatomical systems coevolved. We focused on the primate suborder Strepsirrhini, a living primate group that occupies a very wide range of dietary niches. By making use of both geometric morphometrics and dental topographic analysis, we extend the study of brain-dental ecomorphological evolution beyond measures of size. After controlling for allometry and evolutionary relatedness, differences in brain and dental morphology were found between dietary groups, and brain and dental morphologies were found to covary. Historical trajectories of morphological diversification revealed a strong integration in the rates of brain and dental evolution and similarities in their modes of evolution. Combined, our results reveal an interplay between brain and dental ecomorphological adaptations throughout strepsirrhine evolution that can be linked to diet.

## Introduction

Larger-than-expected brain size (relative to body mass and compared to other mammals) is commonly interpreted as evidence for increased intelligence as an evolutionary novelty in primates [1–3]. Understanding the factors that shaped the evolution of the relatively large primate brain has been a topic of extensive debate, with multiple competing hypotheses postulated over the years [4]. Diet, increased social complexity and the energetic cost of brain tissue development and maintenance have been studied as predictors of brain size in primates [5–7]. Accumulating evidence points to diet as a main predictor of brain size across Primates [4, 8, 9], with factors like sociality playing an important role in certain groups (i.e. anthropoids, [10]). Increasing degrees of frugivory in particular has been identified as a major factor linked to brain enlargement in primates [4]. Frugivory has been interpreted both as a selective pressure (increased spatial information storage and cognitive demands of "extractive foraging";

Vice Principal Academic and Dean to C.L-A. and a NSERC Discovery Grant to M.T.S. The funders had no role in study design, data collection and analysis, decision to publish, or preparation of the manuscript.

**Competing interests:** The authors have declared that no competing interests exist.

[11]) and as an unconstraint (enabling higher energy allocation and turnover; [6]) on brain tissue enlargement.

Studies have discussed the limitations of interpreting brain ecomorphological adaptations solely based on size [12], reflecting the compartmentalised specialisation of the brain and highlighting the need to study brain morphology in a way that also takes shape information into consideration [3, 13, 14]. Understanding how functionally distinct brain structures change differentially in association with ecological (e.g. diet or social complexity) and biological (e.g. allometry) traits can provide a more comprehensive perspective on the eco-evolutionary dynamics of brain evolution [12–15]. Though ecology-driven variations in functionally distinct brain regions and overall brain size may face ontogenetic and functional constraints [16], there is considerable evidence that changes in the morphology of specific brain regions reflect functional adaptions [14, 17] ultimately influencing brain morphology. Parallel lineage-specific and ecology-based patterns of morphological adaptation across brain regions reflect the mosaic evolution of the primate brain as being linked to socioecological differences [14]. Moreover, independent variation patterns between brain size and other phenotypic traits suggest primate brain evolution followed a multi-patterned trajectory [13, 15]. Geometric morphometrics has proved a successful tool at capturing multiple dimensions of brain morphology to reconstruct macroevolutionary processes [13, 15].

Dental morphological variation is also associated with dietary adaptations, providing a potential link between brain and dental ecomorphological evolution [18, 19]. Studies on primate brain-dental functional coevolution have focused on hominins, hypothesising increased dietary quality is associated with a reduction in postcanine dentition and an increase in brain size, resulting in the markedly derived *Homo* large brain and small teeth [18–21]. Recent studies have debated the validity of the inferred coevolution of brain and dental function [18, 19]. Accounting for allometry and evolutionary relatedness reduces the signal of a diet-based brain-dental coevolution across primates, with the exception of prosimians and platyrrhines [18]. Moreover, rates of brain and dental morphological evolution in hominins seem to have followed independent trajectories, indicating decoupled evolution [19]. However, similar to the limitations that come with analysing brain size alone, limiting the study of dental morphological evolution to changes in size neglects other dimensions of dental morphology that have been shown to experience selective pressures [22–24]. Dental topographic analysis has provided novel insights into dental ecomorphological adaptations and evolution, and provides a quantitative toolkit for describing dental shape [25].

Stemming from an inferred frugivorous common ancestor, primates of the suborder Strepsirrhini underwent one of the most impressive adaptive radiations among living primates [26], coupled with significant ecomorphological diversification [25]. The primate suborder Strepsirrhini diverged from haplorrhines approximately 60Ma [26]. Comprising more than 120 living species, they exhibit intermediate relative brain size between non-primate mammals and more derived anthropoids [8, 9], and a variety of dietary and foraging specialisations that are reflected in their dental morphology [25–27]. Strepsirrhines are thought to have diversified in continental Africa, experiencing an early Oligocene partial extinction event, followed by colonisation of Madagascar, perhaps in two independent events [28, 29]. Once in Madagascar, strepsirrhines diversified to fill a range of dietary niches (including folivory, frugivory, gummivory and insectivory) and specialise for different activity periods [25]. A remarkable example of the strepsirrhine diversification is the Aye-Aye (*Daubentonia madagascariensis*) and its combination of morphological traits (e.g. ever-growing incisors, elongated middle digit and squirrel-like skull) associated to its unique percussive foraging strategy [30]. Studies have suggested slower evolutionary rates and a lack of allometry in strepsirrhine primate brain shape [13], and a positive correlation between postcanine teeth size and dietary quality [18].

However, given the highly derived morphology of modern catarrhines (and hominins in particular), it is possible that complex evolutionary processes in Strepsirrhini are obscured when analysing Primates as a whole.

In this study, we investigate the evolution of strepsirrhine brain and dental morphology to test whether diet acted as a link to the integration of brain and dental morphological diversification. To draw a clear link between brain and dental evolution and diet, we account for the effect of allometry and evolutionary relatedness in morphological variation. First, we quantify brain and dental morphology using geometric morphometrics and dental topographic analysis. We decompose brain morphology into its size and shape components, exploring decoupled brain ecomorphological adaptations. After accounting for allometry and phylogenetic structuring, we assess diet-based differences in brain and dental morphologies and test the integration between them. Because diet is closely related to body mass, and also tracks clade membership to some degree, we anticipate that controlling for these factors may remove any effect of diet. Second, we quantify per-dietary group and per-species rates of brain and dental evolution, test for differences between dietary groups and estimate the integration between rates of brain and dental evolution. Finally, we fit competing models of morphological evolution and compare them across dietary groups. Based on the apparently decoupled evolution of brain enlargement and dental reduction in other primate groups that have been assessed, we do not expect to find a strong correlation in evolutionary rates in strepsirrhines.

## Materials and methods

### Data

Data on body mass (BdM), brain mass (BrM) and endocranial volume (ECV) was gathered from the literature at species level, body and brain mass data being averages of mixed-sex samples [4, 8, 9, 31]. Species were assigned to one of three dietary guilds (folivore, frugivore and insectivore), following previous studies on strepsirrhine macroevolution [25, 27]. These categories reflect the predominant component of each species' diet, rather than the level of dietary specialisation. A possible caveat with this approach is the inability to account for the role that gummivory plays in morphological variation, a dietary adaptation that has been associated with morphological specialisation in primates [32]. Phylogenetic relationships were reconstructed following Herrera and Dávalos [26], pruning their phylogeny to match our sample.

### Brain morphology quantification

A dataset of 20 virtual cranial endocasts was obtained from Morphosource. This sample represents 20 species of 20 different genera (87% of generic diversity), covering the phylogenetic and ecological diversity within Strepsirrhini. Only adult specimens were included in our sample. Virtual endocasts were segmented using the 3D imaging software AVIZO® 9.1.1 software. Geometric morphometrics were implemented to quantify brain morphology, decomposing it into its size and shape components. A set of 30 anatomical landmarks were used to capture brain overall morphology (see S1 Table). Of these landmarks, 27 were developed by Bertrand et al. [33] and Ahrens [34]. An additional three were included to capture variation along the brainstem and the caudal-most aspect of the petrosal lobule [35]. In anticipation of future research, these landmarks were chosen based on their ability to capture shape variation across a morphologically diverse group, including other members of Euarchontoglires (i.e. Primates, Dermoptera, Scandentia, Rodentia, Lagomorpha). Landmarks were only placed on the left side of the endocast to avoid covariance between the same points on the left and right side of the endocast. All landmarking was performed in AVIZO® 9.1.1 using a WACOM Cintiq 21UX tablet. A generalised procrustes analyses (GPA) was implemented to control for

methodological artifacts in the landmark data and to decompose brain morphology into its shape (isometry-free) and size (centroid size; CS) components. Linear regression models were used to assess the validity of using CS as a measure of brain size by quantifying the correspondence between it and BrM ($R^2 = 0.955$, $P < 0.001$) and ECV ($R^2 = 0.962$, $P < 0.001$).

## Dental morphology quantification

Three dental topographic metrics were used to quantify dental morphology: Dirichlet Normal Energy (DNE) quantifies changes in curvature across the tooth crown as a proxy for tooth sharpness [36]; Orientation Patch Count Rotated (OPCR) estimates the number of distinct patches on the tooth crown that reflect the topographic complexity of the crown surface [37]; and Relief Index (RFI) measures tooth crown height as a ratio between the crown's 3D and 2D surface areas [38]. Dental topographic metrics per specimen were gathered from one lower second molar, following previous protocols for primate dental topographic analysis [39]. Data for 146 specimens representing the 20 species sampled for brain morphological data was compiled from Fulwood et al. 2021a, 2021b [25, 27] and species averages were used to combine our brain and dental morphological datasets into a single dataset.

## Allometry and phylogenetic structuring

To elucidate diet-driven ecomorphological patterns, we first quantified the effect of allometry and the presence of phylogenetic structuring in our data. Dental, CS and brain shape allometry were tested with a Procrustes regression (using log-transformed BdM), implemented in the procD.lm function in the R package Geomorph [40]. For CS and brain shape, the residuals of the Procrustes regression were used as allometry-corrected data in all further analyses. Phylogenetic structuring in dental morphology, CS and brain shape was assessed by estimating the multivariate K-statistic, using the physignal function in Geomorph with 10,000 iterations for significance testing [40]. The residuals of a phylogenetic linear regression of CS on log-transformed BdM were used as allometry-and-phylogeny-corrected CS and interpreted as a measure of relative brain size, using the procD.pgls function in Geomorph [40]. Given the significant phylogenetic signal and multidimensionality of our dental morphology and brain shape data, we performed a phylogenetic principal component analysis (pPCA) on the correlation matrix of each dataset to control for phylogenetic structuring and reduce the dimensions of our data, using the gm.prcomp function in Geomorph [40]. The first four principal components (each accounting for >10% of variation and in combination accounting for 57.32% of variation) were used for brain shape and the first three principal components (accounting for 100% of variation) were used for dental morphology in all further analyses. Due to our sample size, our per-guild evolutionary modelling analyses limited us to include the first four principal components of brain shape variation.

## Diet-driven morphological variation

We studied whether diet explained patterns of brain and dental morphological variation after controlling for allometry and phylogenetic structuring. Differences in CS, brain shape and dental morphology across dietary guilds were assessed using phylogenetic ANOVAs with 10,000 iterations. Integration between dental and brain morphology was tested using two two-block partial least square analyses, one for CS and another for brain shape, using the two.b.pls function in Geomorph [40].

## Morphological coevolution

We investigated differences in tempo and mode of brain and dental evolution across dietary guilds. To study differences in tempo, we first compared per-group net rates of morphological evolution across dietary guilds for each trait (dental morphology, brain shape and CS), under a Brownian motion model of evolution (BM) with the compare.evol.rates function in Geomorph [40] and 10,000 iterations. Pairwise comparisons of evolutionary rates between dietary guilds were also obtained from the previous analysis. We computed species-level morphological evolutionary rates using the phylogenetic ridge regression method developed in the R package RRphylo [41] for CS, brain shape and dental morphology, separately. We then assessed whether any of the dietary guilds represented a shift in evolutionary regimes, resulting in significantly higher or lower rates of evolution compared to the entire tree. To do this, we used the approach implemented in the search.shift function in RRphylo, using the sparse method and 1,000 iterations for significance testing [41]. We investigated the integration in evolutionary rates between brain and dental morphology using phylogenetic two-block partial least square analysis, accounting for the phylogenetic nonindependence in rates of evolution across species, as implemented in the phylo.integration function in Geomorph [40]. Finally, differences in the mode of evolution across dietary guilds were examined testing competing evolutionary models, using methods developed in the R package mvMORPH [42]. We pooled our data based on dietary guild and pruned our phylogeny to create subtrees for each guild. For each trait (dental morphology, CS and brain shape), we fitted three evolutionary models: BM that assumes a random-walk process, Early Burst (EB) assumes rapid early phenotypic diversification followed by a decline across time (i.e. adaptive radiation) and Ornstein-Uhlenbeck (OU) that assumes evolution under stabilising selection. Sample-size corrected Akaike information criteria (AICc) was used to select the best-supported models, with models with ΔAICc below 2 considered as supported. This procedure was performed for each dietary guild separately.

## Results

Analyses of brain and dental morphological variation show a significant association in their macroevolutionary trajectories driven by dietary adaptations. Significant allometry was only found in brain shape and size, accounting for 11.55% and 93.25% of their variation respectively (Table 1). Statistically significant phylogenetic signal revealed a structuring in brain shape, size and dental morphological variation reflecting evolutionary relatedness (Table 2). After accounting for allometry and phylogenetic structuring, brain shape and dental morphology are significantly different between dietary guilds at $P < 0.03$ and relative brain size nearing significance at $P = 0.07$ (Fig 1 and Table 3). Folivores tended to cluster towards the negative end of pPC1, whereas insectivore species tended to group at the negative end of pPC3, with frugivore species showing the greatest overlap with other guilds (Fig 1A). The Aye-Aye occupied a unique subregion of morphospace, occupying the most positive region along pPC1-3. Comparing brain shape between species at opposite ends of each component revealed antero-posterior and dorsal-ventral distributions of shape variation along pPC1 and pPC2, respectively,

**Table 1. Procrustes linear regressions testing the effect of allometry in brain size and shape and dental morphology.**

|  | Df | SS | MS | R² | F | Z | P |
|---|---|---|---|---|---|---|---|
| Brain shape | 1 | 0.0267 | 0.0267 | 0.1156 | 2.3517 | 2.5053 | **0.0049** |
| Brain size | 1 | 1.8043 | 1.8043 | 0.9325 | 248.8100 | 6.8328 | **0.0001** |
| Dental morphology | 1 | 0.1820 | 0.1820 | 0.0716 | 1.3876 | 0.6537 | 0.2739 |

**Table 2. Tests of phylogenetic signal in brain size and shape and dental morphology.**

|  | K | Z | P |
|---|---|---|---|
| Relative brain size | 0.743 | 1.929 | **0.026** |
| Brain shape | 0.589 | 2.786 | **0.002** |
| Dental morphology | 0.953 | 2.191 | **0.008** |

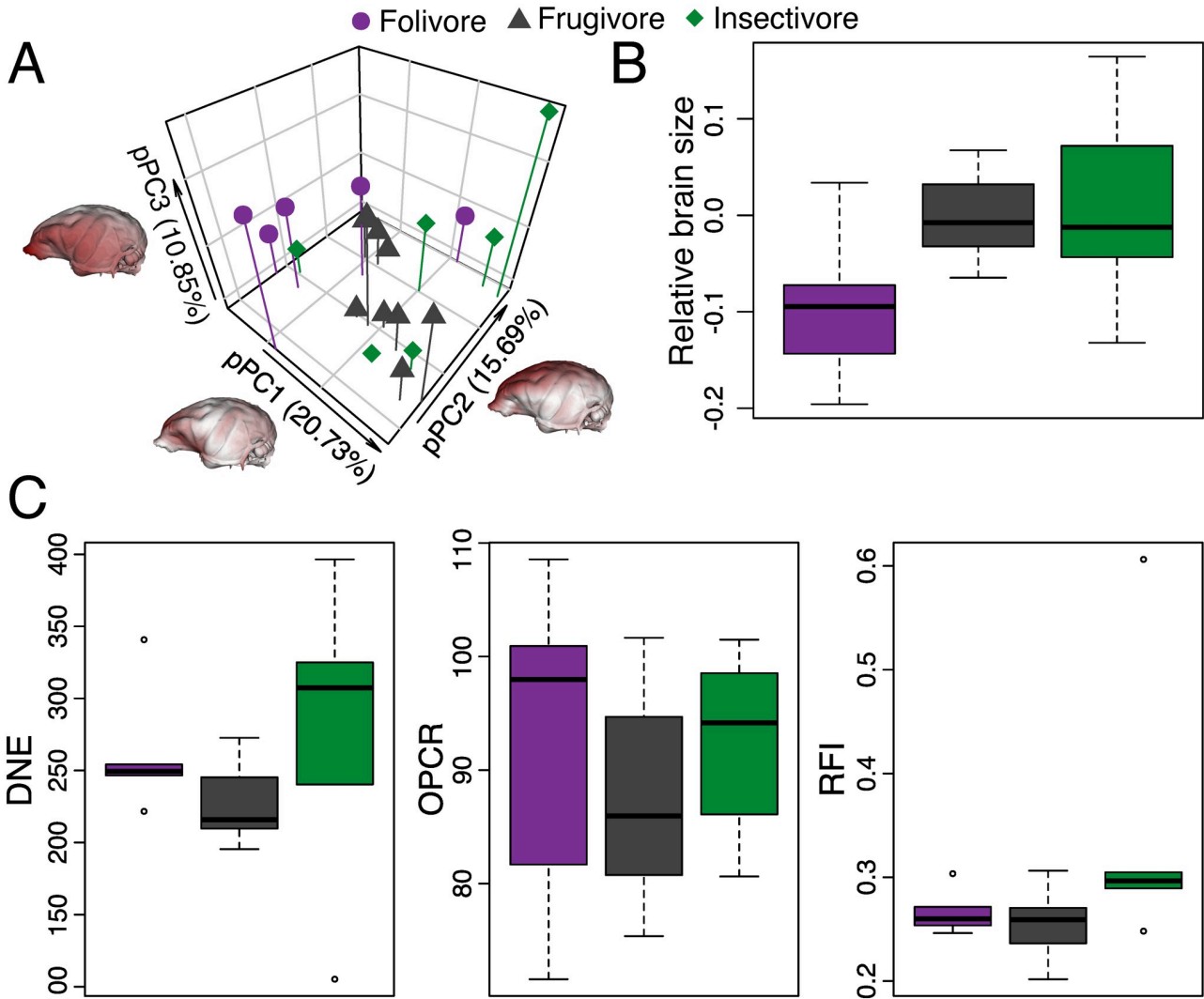

**Fig 1. Patterns of strepsirrhine phenotypic variation across dietary guilds.** Phylomorphospace of allometry-controlled brain shape based on the phylogenetic principal component analysis (A); boxplots of allometry-controlled centroid size (relative brain size) across dietary guilds (C); boxplots of dental morphological variation across dietary guilds (C), representing Dirichlet Normal Energy (DNE; bottom left), Orientation Patch Count Rotated (OPCR; bottom middle) and Relief Index (RFI; bottom right). Endocast heatmaps exemplify differences in brain shape between species occupying opposite ends of each principal component. Heatmaps were obtained by warping the endocast of the modern taxon closest to the inferred average morphology (*Eulemur fulvus*) based on the Procrustes coordinates of species on opposite ends of a principal component and estimating the distance between them.

**Table 3. Procrustes ANOVAs testing for differences in allometry- and phylogeny-corrected brain size and shape and dental morphology across dietary guilds.**

|  | Df | SS | MS | R² | F | Z | P |
|---|---|---|---|---|---|---|---|
| Brain shape | 2 | 0.073 | 0.036 | 0.160 | 1.621 | 2.018 | **0.023** |
| Relative brain size | 2 | 0.035 | 0.018 | 0.264 | 3.055 | 1.430 | 0.075 |
| Dental morphology | 2 | 1.109 | 0.555 | 0.265 | 3.063 | 1.913 | **0.028** |

whereas pPC3 concentrated most shape variation in the frontal lobe. Average relative brain size is markedly lower in folivores (lower than expected for their body mass), less variable in frugivores, and more variable in insectivores (Fig 1B). Dental sharpness (DNE) and crown height (RFI) are lower in frugivore species and higher in insectivore species, whereas average OPCR is higher in folivores and lower in frugivores (Fig 1C). Statistically significant morphological integration between dental morphology and both brain size ($R^2 = 0.493$, $P = 0.028$) and shape ($R^2 = 0.795$, $P = 0.028$) but not between brain size and shape ($R^2 = 0.896$, $P = 0.197$) signal correlated brain-dental variation and decoupled brain size and shape variation.

Per-branch rates of morphological evolution revealed similar patterns in dental morphology and brain shape variation across species, while relative brain size evolution follows an independent pattern (Fig 2A–2C). Interestingly, *D. madagascariensis* was the only species to have high evolutionary rates in all three traits. Frugivore species have lower evolutionary rates across all three traits, showing statistically significant differences in rates of brain shape

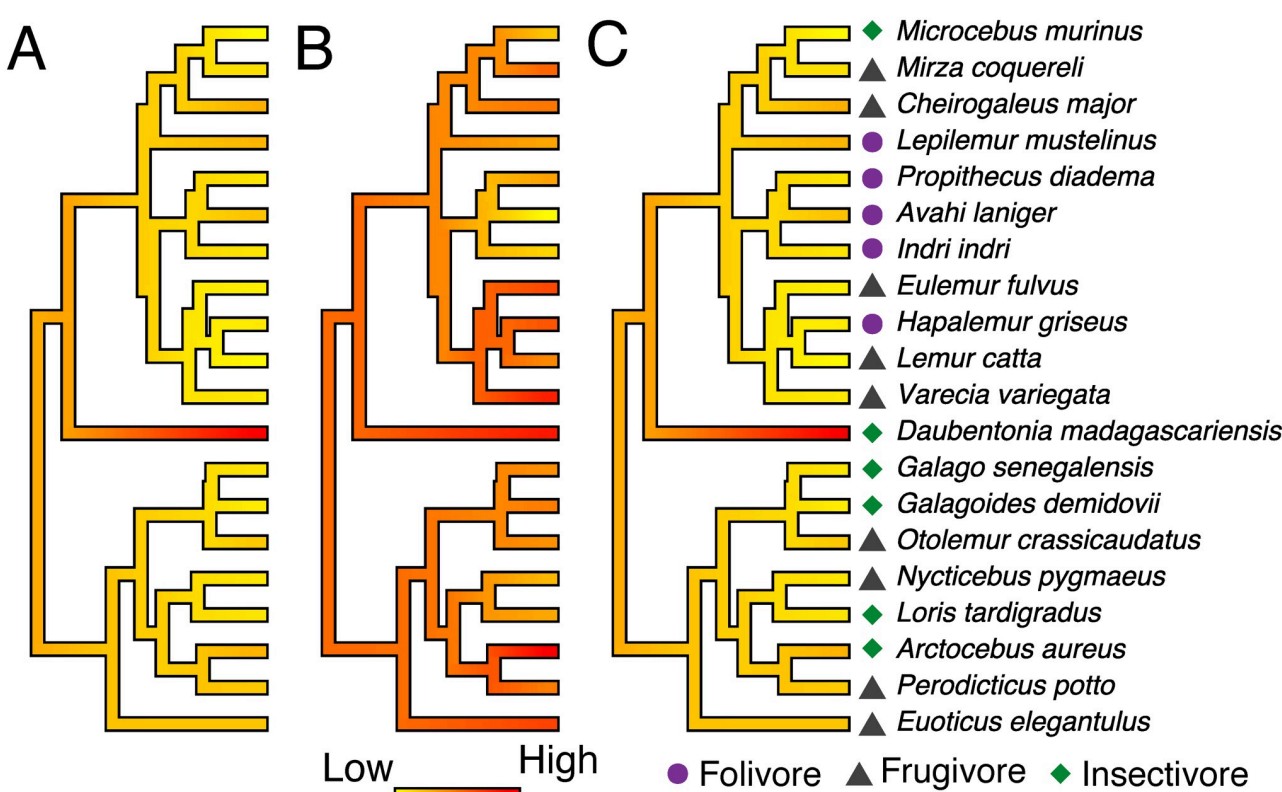

**Fig 2. Per-species rates of phenotypic evolution based on phylogenetic ridge regressions for brain shape (A), brain size (B) and dental morphology (C).** Symbols represent dietary guilds.

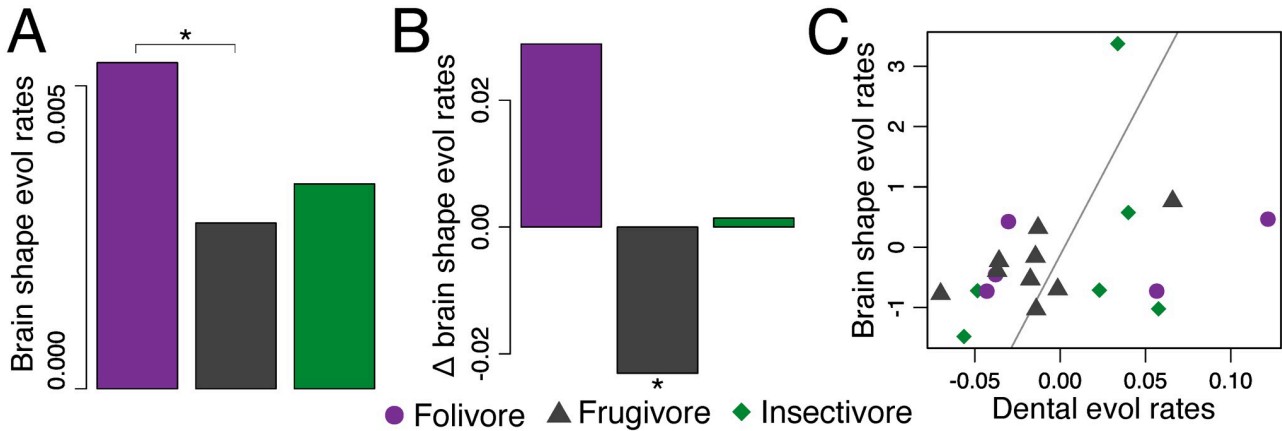

**Fig 3. Comparisons of rates of brain shape evolution across dietary guilds.** Pairwise comparisons of per-guild evolutionary rates in brain shape (A); differences in per-species rates of evolution averaged by guild and compared to the average rate for all strepsirrhines combined (B); partial least square analysis of integration between per-species rates of evolution of brain shape and dental morphology (C). * Represents statistical significance at $P < 0.05$.

evolution (Fig 3 and S2 Table). Comparisons of per-group evolution rates showed frugivores have the lowest net morphological evolution rates in all three traits, albeit only statistically significantly different in brain shape (Fig 3A and 3B, S3 and S4 Tables). Folivores have the highest per-guild and per-species rates of brain size and shape evolution, whereas insectivores have the highest rates of dental morphological evolution (although none was statistically significantly different, see S2–S4 Tables). Significant integration in rates of brain shape and dental morphological evolution (S5 Table), after controlling for the phylogenetic nonindependence in evolutionary rates across species (Table 4), supports the coevolution of brain and dental phenotypes driven by diet (Fig 3C).

Best-supported models of brain and dental morphological evolution differ between dietary guilds (Table 5). Brain shape evolved under stabilising selection (OU model) in folivores and insectivores and following a random-walk (BM) process in frugivores, brain size evolution followed the opposite pattern folivores and insectivores, whereas all three traits evolved under no directional selection (BM) in frugivores. Dental morphology evolved under stabilising selection in folivores and following a BM process in frugivores and insectivores.

## Discussion

Contrary to our predictions, our results show a significant effect of diet on the phenotypic coevolution of the brain and dental morphology of strepsirrhines. Adaptations to both the brain and dentition were crucial factors during the evolution of primates, enabling their ecological diversification [15, 18, 19]. From their early divergence from haplorhines, strepsirrhines evolved remarkably high morphological variability associated with their ecological diversification. Contrary to previous primate macroevolutionary studies that depict

**Table 4. Phylogenetic two-block least square analyses for integration in evolutionary rates between brain size and shape and dental morphology.**

|  | R² | Z | P |
| --- | --- | --- | --- |
| Brain shape_ Relative brain size | 0.107 | 0.465 | 0.665 |
| Brain shape_Dental morphology | 0.493 | 1.968 | **0.028** |
| Brain size_Dental morphology | 0.332 | 1.405 | 0.156 |

**Table 5. Model fitting testing competing evolutionary hypotheses of phenotypic diversification across strepsirrhine dietary guilds.** Three evolutionary models were tested: Brownian motion (BM), early burst (EB) and Ornstein-Uhlenbeck (OU). Models with ΔAICc lower than 2 were inferred as best supported.

| Variable | Guild | Model | LogLik | AICc | ΔAICc |
|---|---|---|---|---|---|
| Relative brain size | Insectivore | BM | 5.116 | -2.232 | **0.000** |
| | | EB | 5.116 | 7.768 | 10.000 |
| | | OU | 5.730 | 6.541 | 8.773 |
| | Folivore | BM | 5.822 | -1.645 | **0.000** |
| | | EB | 5.822 | 18.355 | 20.000 |
| | | OU | 5.829 | 18.342 | 19.987 |
| | Frugivore | BM | 13.640 | -21.279 | **0.000** |
| | | EB | 13.640 | -16.479 | 4.800 |
| | | OU | 15.681 | -20.562 | **0.717** |
| Brain shape | Insectivore | BM | 45.553 | -16.438 | 1234.304 |
| | | EB | 45.153 | -0.307 | 1250.435 |
| | | OU | 49.371 | -1250.742 | **0.000** |
| | Folivore | BM | 49.056 | 13.888 | 311.249 |
| | | EB | 48.708 | 52.584 | 349.945 |
| | | OU | 52.680 | -297.361 | **0.000** |
| | Frugivore | BM | 82.048 | -116.096 | **0.000** |
| | | EB | 80.554 | -107.108 | 8.988 |
| | | OU | 87.956 | -18.821 | 97.275 |
| Dental morphology | Insectivore | BM | -19.597 | 79.693 | **0.000** |
| | | EB | -19.597 | 90.622 | 10.929 |
| | | OU | -16.810 | 303.621 | 223.927 |
| | Folivore | BM | -5.281 | 64.563 | 508.424 |
| | | EB | -5.281 | 85.563 | 529.424 |
| | | OU | -3.069 | -443.861 | **0.000** |
| | Frugivore | BM | -9.070 | 46.728 | **0.000** |
| | | EB | -9.070 | 51.890 | 5.162 |
| | | OU | -2.992 | 79.619 | 32.891 |

Log-likelihood (LogLik), sample-size corrected Akaike information criterion (AICc) and relative fit (ΔAICc) are shown.

strepsirrhine brain evolution as a period of relative stasis [13, 43], our results show a complex multifaceted evolutionary process. After controlling for allometry and phylogenetic structuring, our results showed a significant effect of diet on dental and brain morphology, and a strong integration in the variation and evolutionary trajectories of dental morphology and brain shape, but not brain size. This is in stark contrast with multiple previous studies that have investigated brain evolution only in terms of size [4, 8, 9], and highlights the potential of studying other phenotypic dimensions of brain evolution [3, 14, 15]. Combined, our results provide clear evidence for the effect of diet in the variation and integration of brain and dental morphology during strepsirrhine diversification. Furthermore, our results emphasise the differential effect of factors like diet and sociality have for the evolution of the brain across multiple primate groups [5, 10], highlighting the importance of studying macroevolutionary processes at multiple phylogenetic scales. Given the unique evolutionary trajectory of the Aye-Aye in our results, we replicated our analyses while excluding it to test the sensitivity of our results. All of the notable conclusions hold after removing the Aye-Aye (see S1 File).

Decoupled patterns of brain size and shape variation and evolution in our results support previous findings in hominins [19], platyrrhines [44] and catarrhines [45] that also found

independent patterns of brain size and shape variation. Moreover, our results of brain shape allometry suggest that changes in body mass only explain a fraction of shape variation, as reported in platyrrhine primates [15]. Despite this decoupling of brain size and shape, both traits differed across guilds, indicating independent patterns of adaptation to dietary differences. Significant brain shape allometry in our results contrast with a previous report of non-allometric variation in brain shape in strepsirrhines [13]. The fact that differences in relative brain size between dietary guilds were only nearing significance might be a result of unintended loss of ecological signal after correcting for phylogenetic relatedness, as ecological adaptations also tend to exhibit phylogenetic patterning in primates [46]. Our results of smaller relative brain size in folivores agree with previous findings in strepsirrhine primates [8]. Significant differences in brain shape between guilds also suggest dissimilar adaptations across brain regions depending on different cognitive specialisations, highlighting the organisational complexity of the brain [3, 14]. In strepsirrhines, differences in the relative size of olfactory and visual sensory brain regions have been identified between species with different diets and activity periods (e.g. olfactory structures are enlarged in frugivores; [14]). Compared to other primates, strepsirrhines have relatively enlarged spatial cognition brain areas, which has been associated to their foraging behaviour [14]. Visualisation of brain shape variation between the mean shape of our complete sample and the mean shapes of each dietary guild reveal guild-specific patterns, the insectivore mean shape varying mostly in the dorsal-most aspect of the neocortex, the folivore mean shape primarily in the frontal lobe and the frugivore mean shape varying in a less well-defined pattern across the brain (S1 Fig). While this visualization method identifies shape variation in specific regions of the neocortex (i.e. frontal lobe), additional testing is required to better isolate variation within these subregions with an expanded suite of landmarks specifically tailored to this goal, analyses that are beyond the scope of this study. The present study supports an increasing body of research that points to the need for a better understanding of the modular organisation of the brain [3, 14, 15] and thereby expand our current understanding of primate brain evolution (mainly focused in size). Future studies could apply geometric morphometrics to elucidate integration patterns across brain regions in response to ecological specialisations.

Dental adaptations linked to dietary specialisations in strepsirrhines in our study followed general patterns of dental dietary ecomorphology previously reported in primates [23, 25, 47, 48]. Insectivorous strepsirrhines had higher DNE values, indicating sharper molars adapted to crush and fragment invertebrates' exoskeleton [23]. Similarly, taller tooth crowns (represented by high RFI values) in insectivorous species reflected previous findings indicating decreasing crown height along an animalivory-herbivory gradient in prosimians and platyrrhine primates [23]. Finally, topographic complexity of the molar crown did not reveal a clear pattern discriminating dietary specialisations, although mean OPCR was higher in folivores and lower in frugivores. Ambiguous differentiation of dietary guilds based on OPCR has previously been reported in prosimians, especially after controlling for phylogenetic relatedness [23, 47] Unexpectedly, the highest variation in OPCR was found in folivorous species, which could reflect the importance of specialisations to different folivore niches during the diversification of Malagasy lemurs [49]. Significant integration between dental morphology and brain size and shape suggests that, despite their assumed independence, dietary specialisations represented a major factor canalising their variation to act as a functional unit. Studying the association between brain and dental size in primates, prosimians showed a unique pattern of positive correlation between these two factors and with increased dietary quality [18]. Our results showing positive integration in brain and dental morphology provide evidence that the strepsirrhine brain-dental morphofunctional association extends beyond similarities in size [18].

Our analyses of the mode and tempo of brain and dental phenotypic evolution revealed diet played an important role throughout strepsirrhine evolution, reflecting the general pattern

found for primate brain size evolution [4, 8, 9, 14]. However, we found a consistent pattern of lower evolutionary rates across brain and dental morphological traits associated with frugivory, suggesting that frugivory might not have promoted phenotypic diversification in strepsirrhines. Palaeoecological reconstructions have hypothesised that the strepsirrhine common ancestor was probably a frugivore, from which the clade diversified to specialise in a variety of dietary niches [24, 27]. Lower rates of brain and dental evolution in frugivores in our results would be consistent with frugivory as the ancestral state for strepsirrhines. Moreover, our results of higher evolutionary rates in folivores indicate unexpected phenotypic changes to adapt to folivorous niches. Malagasy colonisation by strepsirrhine primates has been associated with a diversification of folivore niches, linked to significant molecular adaptations to occupy non-overlapping niches and avoid competition [49]. Our results showing accelerated phenotypic changes in folivores can be linked to the folivorous diversification of malagasy lemurs. We hypothesise that previous macroevolutionary studies on the evolution of primate and mammalian brain failed to uncover this unique pattern of folivory-based phenotypic evolution in strepsirrhines because they were being swamped by higher level patterns of change. Evolutionary model fitting revealed that the two most-integrated traits (brain shape and dental morphology) followed the same trajectory, but that trajectory differed across guilds. Brain shape and dental morphology evolved under stabilising selection in folivores and following a BM process in frugivores, signalling the selective pressures acting during the phenotypic evolution of folivorous strepsirrhines and the ancestral frugivorous state in Strepsirrhini [25, 28, 49]. Our results suggest that adapting to insectivory correlated with directional selection in brain shape, the only guild with such a pattern. We hypothesise that unique functional demands associated to insectivores' foraging behaviour are linked to adaptations in specialised brain regions (possibly visual and auditory signal processing regions), rather than an adaptation in whole-brain size or dental morphology [14].

In conclusion, our study provides further evidence of the role diet played during the phenotypic evolution of strepsirrhine primates, revealing clade-specific processes of brain and dental adaptations to folivory and insectivory in one of the earliest primate radiations. In contrast to previous studies in hominins, we show a strong integration between brain and dental phenotypic change even after controlling for allometry and phylogenetic relatedness. Surprisingly, we found an extreme pattern of phenotypic specialisation and accelerated evolution in the Aye-Aye, suggesting its colonisation of Madagascar represented a unique evolutionary event, with a pattern and magnitude of change distinct from those observed in other parts of the strepsirrhine tree that merits further studies [29, 30, 50]. It is possible that our findings of unusual morphological evolution in the Aye-Aye reflect both the relatively recent divergence between the Aye-Aye and its closest relative (*Daubentonia robustus*) during the Pliocene [26], and the extreme morphological convergence between the Aye-Aye and sciurid rodents [30]. Finally, our study reveals the importance of brain shape evolution to understand the diversification of strepsirrhines, as well as primates and vertebrates more generally, shedding light on evolutionary processes otherwise overlooked by studying brain size alone. Future studies should explore the role that gummivory had during the ecomorphological evolution of Strepsirrhini and mammals in general.

## Supporting information

**S1 Table. Description of anatomical landmarks.** Description of anatomical landmarks used to capture brain morphology and analogous landmarks from Bertrand et al. [33] and Ahrens [34].
(DOCX)

**S2 Table. Test for differences in evol rates.** Statistical test for differences in per-species evolutionary rates across dietary guilds and pairwise comparisons.
(DOCX)

**S3 Table. Test for differences in per-guild evol rates.** Statistical test for differences in per-guild evolutionary rates across dietary guilds. Net evolutionary rates are provided per guild for each trait.
(DOCX)

**S4 Table. Pairwise tests of evol rates differences.** Significance values for pairwise statistical tests of differences in per-guild evolutionary rates across traits.
(DOCX)

**S5 Table. Phylogenetic signal in evol rates.** Test of phylogenetic signal in per-species evolutionary rates.
(DOCX)

**S1 Fig. Visualisation of guild-specific brain shape adaptations.** Visualisation of brain shape (Procrustes coordinates) variation between the mean shape of our complete sample and the mean shapes of each dietary guild reveal guild-specific patterns.
(EPS)

**S1 File. Set of replicated results without the Aye-Aye.**
(DOCX)

**S2 File. Raw Landmark data.**
(TXT)

**S3 File. Raw dental topographic data.**
(CSV)

**S4 File. R code to reproduce analyses.**
(R)

## Author Contributions

**Conceptualization:** Camilo López-Aguirre, Madlen M. Lang, Mary T. Silcox.

**Data curation:** Camilo López-Aguirre, Madlen M. Lang.

**Formal analysis:** Camilo López-Aguirre.

**Investigation:** Camilo López-Aguirre.

**Methodology:** Camilo López-Aguirre, Madlen M. Lang.

**Project administration:** Mary T. Silcox.

**Resources:** Mary T. Silcox.

**Software:** Mary T. Silcox.

**Supervision:** Mary T. Silcox.

**Validation:** Madlen M. Lang, Mary T. Silcox.

**Visualization:** Camilo López-Aguirre.

**Writing – original draft:** Camilo López-Aguirre.

**Writing – review & editing:** Madlen M. Lang, Mary T. Silcox.

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
