## [Decision Letter · Decision Letter 0]

11 Apr 2022

PONE-D-22-04796Diet drove brain and dental morphological coevolution in strepsirrhine primatesPLOS ONE

Dear Dr. Lopez-Aguirre,

Thank you for submitting your manuscript to PLOS ONE. After careful consideration, we feel that it has merit but does not fully meet PLOS ONE’s publication criteria as it currently stands. Therefore, we invite you to submit a revised version of the manuscript that addresses the points raised during the review process.

First of all, I want to express to you that I am sorry for the delay in making this return on your work. Finally, we have received the opinions of three reviewers who have made some comments and suggestions, which is why I have decided to request minor revisions.

Basically, I ask you to pay attention mainly to reviewer 2's methodological criticisms and to respond, as usual, whether or not you accept each of the comments and suggestions of the three reviewers. If you are unable to do so or disagree, please justify this in your reply to this editor.

We look forward to receiving your revised manuscript.

Kind regards,

Claudia Patricia Tambussi, Ph.D.

Academic Editor

PLOS ONE

Journal Requirements:

Reviewers' comments:

Reviewer's Responses to Questions

**Comments to the Author**

1. Is the manuscript technically sound, and do the data support the conclusions?

Reviewer #1: Yes

Reviewer #2: Partly

Reviewer #3: Yes

2. Has the statistical analysis been performed appropriately and rigorously? 

Reviewer #1: Yes

Reviewer #2: No

Reviewer #3: Yes

3. Have the authors made all data underlying the findings in their manuscript fully available?

Reviewer #1: Yes

Reviewer #2: No

Reviewer #3: Yes

4. Is the manuscript presented in an intelligible fashion and written in standard English?

Reviewer #1: Yes

Reviewer #2: Yes

Reviewer #3: Yes

5. Review Comments to the Author

Reviewer #1: I appreciated the opportunity to review this manuscript on the coevolution of dental morphology and brain shape in strepsirrhine primates. In this study, the authors examined brain and dental evolution in the context of different dietary guilds (frugivory, folivory, insectivory). They found that brain shape and dental morphology evolved within dietary guilds at similar rates and patterns, suggesting a link between diet and the coevolution of these morphological features. The analyses are appropriately rigorous and the authors’ conclusions are supported by their data (some exceptions noted below, which require further clarification). This study fills a gap in the existing literature on brain/dental evolution in primates, providing a more specific look at this suborder as well as highlighting the importance of examining shape variables in addition to size. In general, I believe this work is suitable for publication in PLOS ONE, pending some revisions. Below, I list some major and minor suggestions for improving the manuscript. I would be happy to review a revised manuscript.

Major:

Page 9: “Species cluster in different subregions of brain morphospace based on dietary guild, with some overlap across guilds where more generalist species group together (Fig. 1).” This feels like an overstatement, based on the considerable overlap in the PCA. Which are the generalist species? They aren’t discussed elsewhere in the manuscript, even though it seems that this could be important context for the dietary guilds. Also, you may wish to address the insectivore outlier in the PCA (Aye-Aye?).

It would be great to include an interpretation of the principal components used for brain shape (Fig. 1); the loadings for these components could be included in a supplementary data file.

Relatedly, on pages 13-14, you address the potential ecological factors driving the evolution of brain shape/cognitive specialization. What kinds of variation in brain shape are associated with the different dietary guilds (in this study, not just in the existing literature)? While I understand that an in-depth analysis of the evolution of brain regions is beyond the scope of this paper, some more detail is warranted here, especially since the following paragraph provides a finer-grained interpretation of DNE, RFI and OPCR values in dental evolution across the three guilds.

Page 11: “Brain shape evolved under stabilising selection (OU model) in folivores and insectivores and following a random-walk (BM) process in frugivores, whereas brain size evolution followed the opposite pattern across guilds.” This statement is confusing in light of Table 5, which indicates that both the BM and OU models have low delta AICc values for frugivore brain size. In fact, the BM model has a lower value, suggesting that brain size does not necessarily follow the opposite pattern as shape across all guilds.

On page 15, you provide the paleoecological context for evolutionary rates in folivores and frugivores; this interpretation/context is missing for rates of insectivore evolution.

Minor:

The introduction would benefit from a smoother transition between the paragraphs on page 4 (from dental topographic analysis to evolutionary history of strepsirrhines). This could be an opportunity to justify your focus on this taxonomic group; why is it a good study system to address the aforementioned questions? The answer is implicit in the following paragraph, but could be better articulated here.

Gummivory is mentioned once in the introduction (page 4) but is not included in the analyses or addressed later in the manuscript. Since you mention it earlier, it might be worth acknowledging and justifying its absence in the study.

I assume that all the data were collected from adult individuals, as age impacts brain/dental size and shape. This should be explicitly stated.

Page 7: What is the justification for including the first four principal components (and not more or fewer) as a proxy for brain shape?

I strongly suggest labeling different panels within each figure with A, B, C, etc., so that you can clearly refer to specific elements of each figure in the text.

I was glad to see you address the Aye-Aye’s extreme accelerated evolution at the very end of the discussion (page 16). This is an interesting result, and an additional sentence speculating about the possible ecological reasons for this pattern would be warranted here.

Reviewer #2: Do I think this paper should be published? No, not really. I will enumerate the many reasons below. However, rather than trying to block this paper from making it into print, I am going to recommend—counter to my own opinion of the work—that this paper is accepted, I would say without revision, but I leave it up to the authors to revise it as they please and resubmit. I promise to accept without further revision, or to work with the authors until either there is something publishable (which I doubt) or they feel they’ve made enough of an effort that this deserves to be in print (its really up to them). First, I will explain why I am taking this counter-intuitive tact. Then I will explain why this paper is a misadventure in P-Hacking, in the attached document.

Reviewer #3: The evolution of primate brain was always of high interest and a matter of debate. Several works have been presented on the subject sometimes differing depending on the primate group analyzed. The present manuscript relating diet to brain and dental morphology coevolution is of remarkable interest due to the diversity and adaptations observed in strepsirrhines, and added important insights to consider for future research on this topic.

The methodology is adequate and was perfectly applied by the authors. Something that may be explained in more detail is the selection of three categories of diet, since it is somewhat difficult to precisely define some categories as frugivorous-folivorous, and that insectivores and folivores may differ in size to be classified. Also, gummivory is a critical category that may lead to unique adaptations. It is suggested to specify why the authors selected those three categories despite others mixed or intermediate to develop the work, although the results were satisfactory by applying the methods.

Among the main results, brain and dental morphology are integrated meaning that diet has major influence to explain their adaptations, differing from some previous studies, and it is notable that especially brain shape and dental morphology are the most integrated traits. These results allow to explore integrations in other primate groups.

Upon some explanations detailed in the text, I recommend publication of this work.

6. PLOS authors have the option to publish the peer review history of their article (what does this mean?). If published, this will include your full peer review and any attached files.

Reviewer #1: No

Reviewer #2: No

Reviewer #3: No

---

## [Author Response · Author response to Decision Letter 0]

2 May 2022

ONE-D-22-04796

Diet drove brain and dental morphological coevolution in strepsirrhine primates

PLOS ONE

Dear Dr. Lopez-Aguirre,

Thank you for submitting your manuscript to PLOS ONE. After careful consideration, we feel that it has merit but does not fully meet PLOS ONE’s publication criteria as it currently stands. Therefore, we invite you to submit a revised version of the manuscript that addresses the points raised during the review process.

First of all, I want to express to you that I am sorry for the delay in making this return on your work. Finally, we have received the opinions of three reviewers who have made some comments and suggestions, which is why I have decided to request minor revisions.

Basically, I ask you to pay attention mainly to reviewer 2's methodological criticisms and to respond, as usual, whether or not you accept each of the comments and suggestions of the three reviewers. If you are unable to do so or disagree, please justify this in your reply to this editor.

Response. We thank the editors and reviewers for their comments and suggestions. Some changes to the manuscript and analyses were made in order to address both reviewers’ comments. Please see responses to specific comments below.

Journal Requirements:

Response: Done. We reviewed the guidelines and edited the manuscript accordingly.

Response: Done. We have revised our data availability statement to specify that all data and code will be available within the supporting information files.

Response: Done.

Review Comments to the Author

Reviewer #1: I appreciated the opportunity to review this manuscript on the coevolution of dental morphology and brain shape in strepsirrhine primates. In this study, the authors examined brain and dental evolution in the context of different dietary guilds (frugivory, folivory, insectivory). They found that brain shape and dental morphology evolved within dietary guilds at similar rates and patterns, suggesting a link between diet and the coevolution of these morphological features. The analyses are appropriately rigorous and the authors’ conclusions are supported by their data (some exceptions noted below, which require further clarification). This study fills a gap in the existing literature on brain/dental evolution in primates, providing a more specific look at this suborder as well as highlighting the importance of examining shape variables in addition to size. In general, I believe this work is suitable for publication in PLOS ONE, pending some revisions. Below, I list some major and minor suggestions for improving the manuscript. I would be happy to review a revised manuscript.

Response: We appreciate the reviewer’s positive comments and constructive suggestions on the manuscript.

Major:

Page 9: “Species cluster in different subregions of brain morphospace based on dietary guild, with some overlap across guilds where more generalist species group together (Fig. 1).” This feels like an overstatement, based on the considerable overlap in the PCA. Which are the generalist species? They aren’t discussed elsewhere in the manuscript, even though it seems that this could be important context for the dietary guilds. Also, you may wish to address the insectivore outlier in the PCA (Aye-Aye?).

It would be great to include an interpretation of the principal components used for brain shape (Fig. 1); the loadings for these components could be included in a supplementary data file.

Response: Done. We have included heatmaps of endocasts comparing the shape of the two species on opposite ends of each principal component, showing where shape change is concentrated. With this, we have replaced our description of the PCA in the Results with a more accurate and informative section. See page 9. We avoid discussing the loadings of these components as they are rotated based on phylogenetic relatedness, making it impossible to infer detailed morphological changes from them. 

Relatedly, on pages 13-14, you address the potential ecological factors driving the evolution of brain shape/cognitive specialization. What kinds of variation in brain shape are associated with the different dietary guilds (in this study, not just in the existing literature)? While I understand that an in-depth analysis of the evolution of brain regions is beyond the scope of this paper, some more detail is warranted here, especially since the following paragraph provides a finer-grained interpretation of DNE, RFI and OPCR values in dental evolution across the three guilds.

Response: Done. We have included a supplementary figure visualising shape variation between the mean shape of our complete sample and the mean shape of each dietary guild, and briefly discussed guild-specific patterns in the discussion. As the reviewer said, an in-depth analysis of guild-specific brain adaptations is beyond the scope of the current study, so we’ve tried to provide additional information without venturing into speculative arguments. See pages 14-15 and S1 Fig. 

Page 11: “Brain shape evolved under stabilising selection (OU model) in folivores and insectivores and following a random-walk (BM) process in frugivores, whereas brain size evolution followed the opposite pattern across guilds.” This statement is confusing in light of Table 5, which indicates that both the BM and OU models have low delta AICc values for frugivore brain size. In fact, the BM model has a lower value, suggesting that brain size does not necessarily follow the opposite pattern as shape across all guilds.

Response. We thank the reviewer for highlighting this inconsistency. We have amended this sentence to better reflect the different patterns across guilds and how all three traits seemed to have evolved under no directional selection (BM) in frugivores. See page 11.

On page 15, you provide the paleoecological context for evolutionary rates in folivores and frugivores; this interpretation/context is missing for rates of insectivore evolution.

Response. Done: We have included two additional sentences in page 16 specifically discussing our findings with respect to insectivory and strepsirrhine evolution. We argue that our results indicate insectivory followed a unique evolutionary trajectory where it only correlated with directional selection in brain shape. Moreover, we hypothesise that unique functional demands associated with insectivores’ foraging behaviour are linked to adaptations in specialised brain regions (possibly visual and auditory signal processing regions), rather than an adaptation in whole-brain size or dental morphology. 

Minor:

The introduction would benefit from a smoother transition between the paragraphs on page 4 (from dental topographic analysis to evolutionary history of strepsirrhines). This could be an opportunity to justify your focus on this taxonomic group; why is it a good study system to address the aforementioned questions? The answer is implicit in the following paragraph, but could be better articulated here.

Response. Done: We have included an additional sentence in-between the two paragraphs in question to smooth the transition of topics and better highlight the importance of focusing on studying strepsirrihine primates. See page 4.

Gummivory is mentioned once in the introduction (page 4) but is not included in the analyses or addressed later in the manuscript. Since you mention it earlier, it might be worth acknowledging and justifying its absence in the study.

Response: Done. Following suggestions made by reviewers 1 and 3, we have better clarified our dietary categories and clearly stated the caveat in our classification and the importance of exploring the effect of gummivory on the ecomorphological evolution of Strepsirrhini. See pages 6 and 16.

I assume that all the data were collected from adult individuals, as age impacts brain/dental size and shape. This should be explicitly stated.

Response: Done. We have added additional information clarifying that all specimens analysed were adults. See page 6.

Page 7: What is the justification for including the first four principal components (and not more or fewer) as a proxy for brain shape?

Response: Done. We have included additional information in the methods to better explain our reasoning for this (see page 7). Our per-guild sample size limited our capacity to include more principal components in the evolutionary modelling analyses, so we decided to include: 1) all the components that each explained at least 10% of variation and 2) as many as the modelling analyses could take.

I strongly suggest labeling different panels within each figure with A, B, C, etc., so that you can clearly refer to specific elements of each figure in the text.

Response: Done. We have labelled different panels within each figure and amended the results and legends accordingly. 

I was glad to see you address the Aye-Aye’s extreme accelerated evolution at the very end of the discussion (page 16). This is an interesting result, and an additional sentence speculating about the possible ecological reasons for this pattern would be warranted here.

Response: Done. We have included an additional sentence interpreting our results as possible evidence of the relatively recent divergence between the Aye-Aye and its closest relative (D. robustus) in the Pliocene, and the extreme morphological convergence between the Aye-Aye and sciurid rodents. See page 16.

Reviewer #2: Do I think this paper should be published? No, not really. I will enumerate the many reasons below. However, rather than trying to block this paper from making it into print, I am going to recommend—counter to my own opinion of the work—that this paper is accepted, I would say without revision, but I leave it up to the authors to revise it as they please and resubmit. I promise to accept without further revision, or to work with the authors until either there is something publishable (which I doubt) or they feel they’ve made enough of an effort that this deserves to be in print (its really up to them). First, I will explain why I am taking this counter-intuitive tact. Then I will explain why this paper is a misadventure in P-Hacking, in the attached document.

Response. The reviewer has provided a detailed review, providing in-depth opinions on a variety of topics and raising a myriad of issues. Unfortunately, the tone the reviewer used when writing the revision muddies the criticisms they are trying to convey, which led the reviewer to extensively discuss things ranging from the peer-review process generally and scientific publishing as a whole, to questioning our work ethic and desire to do high-quality science. Confusingly, the reviewer concludes promising “to accept without further revision” while at the same time promising “to work with the authors until … there is something publishable (which I doubt)”. Based on this comment, we fail to see how the reviewer is interested in engaging in a constructive and professional scientific exchange of ideas. Despite these differences, we have gone through the reviewer’s comments and will provide a point-by-point response to the main issues. First, we respectfully disagree with the reviewer’s overall assessment of our study, but especially with the tone and the intentions that seem to have motivated this review. We might share some of the reviewer’s concerns in terms of the peer-review process and scientific publishing, but contrary to the reviewer we don’t see this as a productive setting to discuss this. In terms of the specific criticisms to our study, we are addressing the main points:

1) P-Hacking: The reviewer argues that our study is “a misadventure in P-Hacking”. The reviewer employs a series of interpretations of aspects of our study to justify why they’re certain that we engaged in P-hacking in order to find “significant results”. Unsurprisingly, we strongly disagree with the reviewer on this point for several reasons; 1) The reviewer states that “the premise and conclusions of this paper are not controversial”, which in our opinion seems counterintuitive to the need for P-Hacking. If our results are so underwhelming and predictable, why would we need to doctor our data or statistical analyses to find an already predictable and expected pattern? 2) We clearly stated our reasoning and expectations for our results in the manuscript, and as we clearly stated in our discussion, our results were contrary to our expectations in some ways. We followed a structured and sequential line of observation, questioning and testing, starting with a premise and when our results didn’t align with it, we reoriented our discussion to try and explain the unexpected patterns we found. Our counterargument would be, how could we have engaged in P-hacking when some of our results go against our expectations and predictions? 3) The reviewer also states that we implemented a “strange and unnecessary over-application of various analytic tools”, while at the same time criticising the ability of individual analyses to test our hypotheses. We are aware of the different limitations some of our analyses have, which is precisely the reason why we decided to use a combination of tests to thoroughly assess our data. Rather than relying on a single test with limited capacity to investigate our data, we strove to compensate for the limitations that come with each test. 

2) The effect of the Aye-Aye: The reviewer repeatedly argued in multiple occasions that our results were driven by the unusual biology of the Aye-Aye and that without it, all our findings wouldn’t hold. The reviewer also argues that we should just not analyse the Aye-Aye because it’s a “biological outlier”. We respectfully disagree with the premise that one can simply deem a species a “biological outlier” and ignore it. Ignoring the presence of species with unique adaptations that represent rare evolutionary processes artificially homogenises and standardises nature. Having said that, we agree with the reviewer that it is important to test the sensitivity of our results to the presence of the Aye-Aye, so we have included an additional set of analyses excluding the Aye-Aye. The results of these additional analyses show that our results are not driven by this one species’ unusual evolutionary trajectory (see S1 File with five supplementary tables of results). 

3) Allometry: The reviewer disagrees with our usage of residuals as an allometry-corrected trait. 1) Contrary to the reviewer’s argument, not all phylogenetic comparative methods allow the inclusion of confounding factors (e.g. comparing evolutionary rates across groups while accounting for allometry), this is the reason why it is common practice to use the residuals in these kind of analyses, and 2) The reviewer’s assumption that allometric trajectories have to vary across taxa (which is true but not a universal pattern, as has been shown across several groups) and that this renders the use of residuals non-informative would be relevant to a different research question altogether, i.e., how allometric trajectories may vary across taxa. That is not the focus of this study, so we argue that the use of the residuals of a single allometric trajectory for our sample is valid, as it is only used to test and account for the effect of size in our sample.

4) Accounting for phylogenetic signal: The reviewer criticises our usage of the components of a phyloPCA to obtain data that accounts for phylogenetic relatedness for its use in subsequent phylogenetic analyses. Since what we are testing here is the effect of a single ecological trait (diet) on the evolution of brain and dental morphology, not accounting for the possible effect of phylogenetic relatedness on morphological traits would make it impossible to truly discern the effect of diet. Moreover, controlling for phylogenetic relatedness in the morphological data doesn’t erase the phylogenetic information in our analyses, because that is explicitly incorporated by way of the phylogeny that is used as a scaffold for all analyses. Additionally, these methods cannot account for the possible phylogenetic structuring in our morphological data while at the same time reconstructing their evolution based on biotic or abiotic traits (explicitly untangling whether morphological variation is linked to evolutionary relatedness or ecological factors). Therefore, it is common practice to use ordination analyses to reduce the dimensionality of the data while accounting for relatedness, in order to test the effect of an ecological or biological factor like diet. Not using our approach would limit our results to reconstructing macroevolutionary trajectories, without investigating mechanistic models like the effect of diet on such trajectories.

5) Data and code: Contrary to the reviewer’s assertion that we are “withholding” the data, we clearly stated in our data availability statement that all raw data and code will be made freely accessible. In this revision we are including our raw data and code in the supplements.

Reviewer #3: The evolution of primate brain was always of high interest and a matter of debate. Several works have been presented on the subject sometimes differing depending on the primate group analyzed. The present manuscript relating diet to brain and dental morphology coevolution is of remarkable interest due to the diversity and adaptations observed in strepsirrhines, and added important insights to consider for future research on this topic.

The methodology is adequate and was perfectly applied by the authors. Something that may be explained in more detail is the selection of three categories of diet, since it is somewhat difficult to precisely define some categories as frugivorous-folivorous, and that insectivores and folivores may differ in size to be classified. Also, gummivory is a critical category that may lead to unique adaptations. It is suggested to specify why the authors selected those three categories despite others mixed or intermediate to develop the work, although the results were satisfactory by applying the methods.

Among the main results, brain and dental morphology are integrated meaning that diet has major influence to explain their adaptations, differing from some previous studies, and it is notable that especially brain shape and dental morphology are the most integrated traits. These results allow to explore integrations in other primate groups.

Response: We appreciate the reviewer’s positive comments and constructive suggestions on the manuscript.

Upon some explanations detailed in the text, I recommend publication of this work.

Response: Done. Following suggestions made by reviewers 1 and 3, we have better clarified our dietary categories and the importance of exploring the effect of gummivory on the ecomorphological evolution of Strepsirrhini. See pages 6 and 16.

---

## [Editor Report · Decision Letter 1]

13 May 2022

Diet drove brain and dental morphological coevolution in strepsirrhine primates

PONE-D-22-04796R1

Dear Dr. Camilo López-Aguirre,

We’re pleased to inform you that your manuscript has been judged scientifically suitable for publication and will be formally **accepted **for publication once it meets all outstanding technical requirements.

Kind regards,

Claudia Patricia Tambussi, Ph.D.

Academic Editor

PLOS ONE

Additional Editor Comments 

I appreciate that you have evaluated all the opinions expressed by the three reviewers, focusing on those specifically related to the contents of this work. The main questions raised by them, especially reviewer 1 and 3 have been satisfactorily considered in this new version of the manuscript.

---

## [Editor Report · Acceptance letter]

27 May 2022

PONE-D-22-04796R1 

Diet drove brain and dental morphological coevolution in strepsirrhine primates 

Dear Dr. Lopez-Aguirre:

I'm pleased to inform you that your manuscript has been deemed suitable for publication in PLOS ONE. Congratulations! Your manuscript is now with our production department. 

Kind regards, 

on behalf of

Dr. Claudia Patricia Tambussi 

Academic Editor

PLOS ONE